# Automated Intraoperative Lumpectomy Margin Detection using SAM-Incorporated Forward-Forward Contrastive Learning

**Tyler Ward**[1]  iD                                      TYLER.WARD@UKY.EDU

**Braxton McFarland**[1]  iD                    BRAXTONJMCFARLAND@GMAIL.COM

**Sahar Nozad**[1]                                     SAHAR.NOZAD@UKY.EDU

**Talal Arshad**[1]                                          T.ARSHAD@UKY.EDU

**Hafsa Nebbache**[1]                            HAFSA.NEBBACHE@UKY.EDU

**Jin Chen**[2]  iD                                              JINCHEN@UAB.EDU

**Xiaoqin Wang**[1]                                  XIAOQIN.WANG@UKY.EDU

**Abdullah Imran**[1]  iD                                    AIMRAN@UKY.EDU

[1] *University of Kentucky, Lexington, KY 40506, USA*

[2] *University of Alabama at Birmingham, AL 35233, USA*

## Abstract

Complete removal of cancerous tumors with a negative specimen margin during lumpectomy is essential to reduce breast cancer recurrence. However, interpretation of 2D specimen radiography (SR) by radiologists, the current method used to assess intraoperative margin status, has limited accuracy. This study aims to improve positive margin detection performance on SRs by leveraging cutting-edge deep learning models. We propose a novel lumpectomy margin assessment method using an innovative integration of a contrastive pre-training strategy (Forward-Forward Contrastive Learning) and a segmentation foundation model (Segment Anything Model). Experimental results on an independently annotated breast SRs demonstrate the effectiveness of the proposed method in accurately detecting positive margins on SRs.

**Keywords:** Breast, Lumpectomy, Specimen radiography, Cancer margin, Forward-Forward, Contrastive learning, SAM

## 1. Introduction

Accurate intraoperative detection of positive margins during breast-conserving surgery (BCS) is critical to minimizing cancer recurrence and avoiding repeat surgeries (Siegel et al., 2025). Radiography-based inspection, the standard practice during BCS for margin assessment, offers limited sensitivity and often fails to differentiate between tumor tissue and surrounding structures reliably (Scimone et al., 2021). This leads to roughly 20–25% of patients requiring re-excision (Scimone et al., 2021).

Recent advancements in deep learning have shown promise in medical imaging, but most margin assessment approaches rely on supervised learning and require extensive annotation, which is labor-intensive and difficult to scale (To et al., 2023). Moreover, foundation models such as the Segment Anything Model (SAM) (Kirillov et al., 2023; Ravi et al., 2024) are not tailored to specimen radiographs (SRs) and typically require manual prompting.

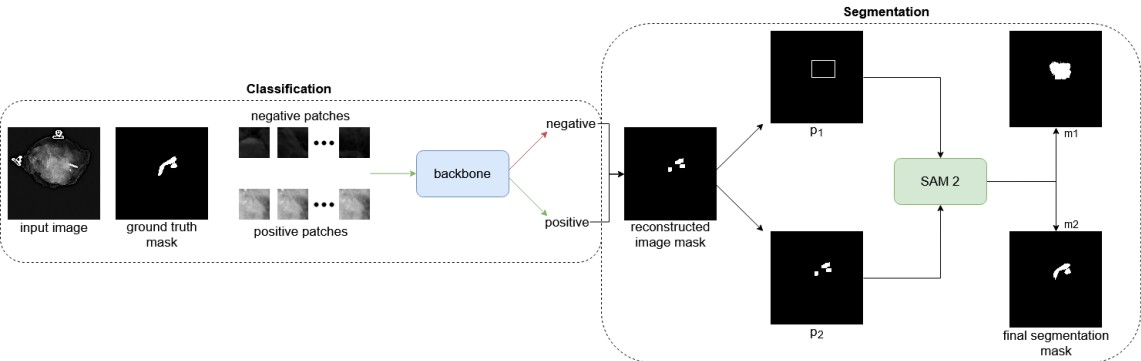

Figure 1: Illustration of the proposed breast cancer margin detection workflow: Initial image mask is reconstructed from the patch-level margin detection outputs and fed to SAM 2 for generating the final segmentation.

In this work, we propose FFCL-SAM, a novel method that combines self-supervised Forward-Forward Contrastive Learning (FFCL) (Ahamed et al., 2023) with SAM for automated, accurate lumpectomy margin detection. Our method first classifies radiograph patches using an FFCL-pretrained classification model and then uses the predictions to reconstruct masks from the patches and train SAM under a few-shot setting for refined pixel-level segmentation (see Fig. 1). This integrated approach improves detection performance and may reduce re-excision rates, with fast inference suitable for intraoperative use.

## 2. Methods

FFCL employs the Forward-Forward Algorithm (FFA) (Hinton, 2022) and performs two-stage pre-training (local followed by global contrastive learning) without requiring any back-propagation (Ahamed et al., 2023, 2024). To generate initial segmentation masks from FFCL-based classification, we reconstruct images from their patches in the form of a binary mask, where an activation in the mask indicates a positive prediction, and no activation indicates the opposite. To improve the accuracy of these coarse reconstructed masks, we refine them using SAM 2, which we fine-tune using a small labeled set of SRs to aid better approximation of the ground truth margin annotations. Once fine-tuned, SAM 2 receives two prompts based on the class predictions from the FFCL pipeline: a bounding box encompassing the predicted region and the coarse binary mask itself, allowing the model to leverage both spatial and form information when making its prediction.

## 3. Experiments and Results

In this HIPAA (Health Insurance Portability and Accountability Act)-compliant retrospective study, intraoperative SRs were collected from 215 lumpectomy procedures performed between 2019 and 2022 at the University of Kentucky Markey Cancer Center Comprehensive Breast Care Center. The Institutional Review Board (IRB) has approved, and all images

Table 1: Classification (top) and segmentation (bottom) performance. Best scores are **bolded** and second best are underlined.

| Classification Results | | | | | |
|---|---|---|---|---|---|
| Model | Accuracy | F1 | Precision | Recall | AUC |
| ConvNeXt w/ FFCL | $0.7510 \pm 0.0029$ | $0.6396 \pm 0.0100$ | $0.6542 \pm 0.0186$ | $0.6322 \pm 0.0072$ | $0.7751 \pm 0.0002$ |
| ViT w/ FFCL | **$0.7816 \pm 0.0316$** | $0.6898 \pm 0.0373$ | **$0.7882 \pm 0.0582$** | $0.6006 \pm 0.0387$ | $0.8067 \pm 0.0530$ |
| ResNet-18 w/ FFCL | $0.7815 \pm 0.0174$ | **$0.6914 \pm 0.0238$** | $0.7478 \pm 0.0656$ | **$0.6900 \pm 0.0142$** | **$0.8455 \pm 0.0152$** |

| Segmentation Results | | | | | | | | |
|---|---|---|---|---|---|---|---|---|
| Mask | Positive Margin | | | Negative Margin | | | Overall | | |
| | DSC | HD | Accuracy | DSC | HD | Accuracy | DSC | HD | Accuracy |
| Reconstructed Mask | 0.6313 | 85.3282 | 0.8157 | 0.6861 | 179.5214 | 0.8431 | 0.6587 | 132.4248 | 0.8294 |
| SAM-zero shot | 0.5206 | 87.4305 | 0.7603 | 0.7631 | 307.6746 | 0.8816 | 0.6419 | 197.5535 | 0.8210 |
| SAM2-zero-shot | 0.8861 | 6.8267 | 0.9431 | 0.9220 | 9.3258 | 0.9610 | 0.9041 | 8.0763 | 0.9521 |
| SAM2-few-shot | **0.9053** | **6.6736** | **0.9527** | **0.9443** | **9.0721** | **0.9722** | **0.9248** | **7.8729** | **0.9625** |

have been de-identified before use. Samples were excluded if they lacked anatomic orientation, intraoperative labeling, or pathology-confirmed malignancy. Tumor localization and 3D orientation were confirmed using diagnostic mammograms and post-biopsy imaging. Each SR was annotated with three region labels: nonmalignant tissue, tumor presence, and pathology-confirmed positive margin. The final dataset included 92 SR images from 46 patients split into training, validation, and test sets (38/2/6).

For the classification task, each SR was divided into overlapping 64×64 patches with a stride of 3. To address class imbalance, fewer patches were extracted from negative regions. For validation and testing, patches from positive margin regions only were used to better assess generalization. Five SRs with corresponding ground-truth masks were used to fine-tune SAM 2 in a few-shot setting. The coarse masks reconstructed from patch-level predictions were refined using SAM 2, which received both bounding box and mask prompts derived from the FFCL predictions. Table 1 summarizes the performance of the classification and segmentation approaches. Among the FFCL-pretrained backbones, ResNet-18 has the best classification performance, with top scores in F1, recall, and AUC, and second-best accuracy and precision. For segmentation, zero-shot SAM 2 already shows strong performance over the coarse baseline, but few-shot fine-tuning yields the best results overall on Dice similarity coefficient (DSC), Hausdorff distance (HD), and accuracy metrics.

## 4. Conclusions

FFCL-SAM achieves high classification performance for detecting positive margins in SRs. Evaluation of the subsequent segmentation task demonstrates that SAM 2 under a few-shot training setting shows promise in refining segmentation masks, improving in performance compared to the zero-shot settings. The findings from this study indicate the potential to reduce the rate of lumpectomy re-excision and improve patient outcomes through the use of deep learning methods, particularly the integration of domain-specific pre-training with a generic foundation model.

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
