# OpenReview forum: "Automated Intraoperative Lumpectomy Margin Detection using SAM-Incorporated Forward-Forward Contrastive Learning"
_MIDL.io/2025/Short_Papers — MIDL 2025 - Short Papers_

### Official Review · Reviewer_2iUx · 2025-04-29

**Rating:** 4
**Confidence:** 4

**Summary:**

The manuscript introduces a novel and clinically relevant deep-learning approach that effectively addresses breast cancer treatment challenges. It demonstrates innovation by integrating Forward-Forward Contrastive Learning (FFCL) with Segment Anything Model, and preliminary results show impressive performance metrics in classification and segmentation tasks.

**Strengths:**

Innovative combination of methodologies with clear clinical significance.

Strong experimental results indicate notable performance improvements.

**Weaknesses:**

Dataset size and single-center validation may limit its generalizability.

Limited exploration of computational efficiency and robustness via ablation studies.

While the few-shot setting improves segmentation results, this approach still requires manual annotation for fine-tuning, which might be challenging in clinical environments with limited resources or expertise.

Provide a brief sensitivity analysis or more detailed justification of method components (FFCL and SAM2 individually).

---

### Decision · Program_Chairs · 2025-05-01

Accept